# The Interdependency and Co-Regulation of the Vitamin D and Cholesterol Metabolism

**DOI:** 10.3390/cells10082007

**Published:** 2021-08-06

**Authors:** Tara Warren, Roisin McAllister, Amy Morgan, Taranjit Singh Rai, Victoria McGilligan, Matthew Ennis, Christopher Page, Catriona Kelly, Aaron Peace, Bernard M. Corfe, Mark Mc Auley, Steven Watterson

**Affiliations:** 1Northern Ireland Centre for Stratified Medicine, C-TRIC, Altnagelvin Hospital Campus, School of Biomedical Sciences, Ulster University, Derry BT47 6SB, UK; tarawarren96@hotmail.co.uk (T.W.); rm.mcallister@ulster.ac.uk (R.M.); t.rai@ulster.ac.uk (T.S.R.); v.mcgilligan@ulster.ac.uk (V.M.); Ennis-M4@ulster.ac.uk (M.E.); Page-C7@ulster.ac.uk (C.P.); c.kelly@ulster.ac.uk (C.K.); 2Department of Chemical Engineering, Faculty of Science & Engineering, University of Chester, Parkgate Road, Chester CH1 4BJ, UK; amy.morgan@chester.ac.uk (A.M.); m.mcauley@chester.ac.uk (M.M.A.); 3Cardiology Unit, Western Health and Social Care Trust, Altnagelvin Regional Hospital, Derry BT47 6SB, UK; Aaron.Peace@westerntrust.hscni.net; 4Human Nutrition Research Centre, Institute of Cellular Medicine, William Leech Building, Medical School, Newcastle University, Framlington Place, Newcastle upon Tyne NE2 4HH, UK; Bernard.Corfe@newcastle.ac.uk

**Keywords:** cholesterol, vitamin D, systems biology, regulation, pathway biology

## Abstract

Vitamin D and cholesterol metabolism overlap significantly in the pathways that contribute to their biosynthesis. However, our understanding of their independent and co-regulation is limited. Cardiovascular disease is the leading cause of death globally and atherosclerosis, the pathology associated with elevated cholesterol, is the leading cause of cardiovascular disease. It is therefore important to understand vitamin D metabolism as a contributory factor. From the literature, we compile evidence of how these systems interact, relating the understanding of the molecular mechanisms involved to the results from observational studies. We also present the first systems biology pathway map of the joint cholesterol and vitamin D metabolisms made available using the Systems Biology Graphical Notation (SBGN) Markup Language (SBGNML). It is shown that the relationship between vitamin D supplementation, total cholesterol, and LDL-C status, and between latitude, vitamin D, and cholesterol status are consistent with our knowledge of molecular mechanisms. We also highlight the results that cannot be explained with our current knowledge of molecular mechanisms: (i) vitamin D supplementation mitigates the side-effects of statin therapy; (ii) statin therapy does not impact upon vitamin D status; and critically (iii) vitamin D supplementation does not improve cardiovascular outcomes, despite improving cardiovascular risk factors. For (iii), we present a hypothesis, based on observations in the literature, that describes how vitamin D regulates the balance between cellular and plasma cholesterol. Answering these questions will create significant opportunities for advancement in our understanding of cardiovascular health.

## 1. Introduction

Interest in vitamin D has expanded significantly in recent years with the number of publications featuring the fat-soluble vitamin growing rapidly. Vitamin D has a crucial role in skeletal health, affecting bone mineralisation and calcium and phosphate homeostasis along with the regulation of the parathyroid hormone [1,2]. Traditionally, vitamin D deficiency is associated with the pathogenesis of rickets and osteomalacia, affecting children and adults, respectively [3,4]. However, more recently, the dysregulation of vitamin D has been implicated in a range of immunological conditions including cancer [5], respiratory conditions [6], rheumatoid arthritis [7], and diabetes [8], underscoring its important role as an immunomodulator [9]. Hence, recent studies have investigated the efficacy of vitamin D supplementation as a treatment strategy across a multitude of conditions [10,11].

Intriguingly, vitamin D is inextricably linked with cholesterol metabolism with the two metabolisms sharing an extensive common biosynthesis pathway. Cholesterol is a lipid with many roles. It is a vital component of cellular membranes [12], a precursor to bile acids, steroids, and oxysterols [13], and is implicated in neurological development [14], cardiovascular health [15,16], innate immunity [17,18], and gallbladder disease [19]. Importantly, its dysregulation can result in elevated total blood cholesterol, and LDL-C, which have been associated with cardiovascular risk [16,20]. The interplay between the two metabolic pathways is complex. For instance, vitamin D deficiency has been associated with increased incidence of CVD [21,22] and vitamin D supplementation has been related to an improvement in atherogenic lipid markers [23]. However, many studies have reported that cardiovascular events were ultimately unaffected by supplementation [24,25,26]. Even more intriguingly, evidence suggests that statin therapy, used in the treatment of hypercholesterolaemia, does not influence the plasma levels of vitamin D [27]. This is counter-intuitive, as statins inhibit a key mechanism in the biosynthesis pathway shared by both metabolisms. Moreover, it has been observed that vitamin D supplementation reduces the side effects associated with statin treatment [28], reduces the concentration of the statin and its metabolites, whilst conversely enhancing the action of statins [29]. Additionally, it is clear that dysregulation of cholesterol and vitamin D metabolism occurs with age and that dysregulation correlates with a rise in age-associated diseases [30,31], but it is poorly understood how this dysregulation develops. 

The interplay between these two elaborate metabolic pathways can be best described with the use of systems biology, which enables their complex biochemical interactions to be viewed in an integrated manner. In this paper, we begin by exploring the known role of cholesterol in health and disease and then the known role of vitamin D in health and disease. Next, the bidirectional relationship between cholesterol and vitamin D metabolisms is outlined before we discuss the role of statins, the feedback mechanisms at play, and the role of known and mutations. Following this, the role of systems biology is emphasised by compiling a detailed review of the computational models of the two pathways. Finally, we describe the development of a new systems biology network diagram that underpins these complex metabolic pathways. This allows us to explore experimental results and observations in the context of systems level behaviour and ultimately to identify which results are consistent with our systems level understanding and which results are in contradiction. 

## 2. Cholesterol in Health and Disease 

Cholesterol is absorbed from the diet and synthesized in cellular pathways with biosynthesis contributing approximately 80% of serum cholesterol [32]. The liver plays a central role in synthesizing and regulating cholesterol biosynthesis [33,34]. Blood cholesterol is determined by multiple factors: biosynthesis, dietary intake, absorption, cellular uptake, cellular efflux, excretion metabolism to bile acids [35]. It is well established that elevated blood cholesterol is associated with increased cardiovascular risk. This is due to the role that cholesterol plays in the pathogenesis of atherosclerosis. In this process, low density lipoproteins, containing triglycerides and carry cholesterol, form low density lipoprotein cholesterol (LDL-C) particles that undergo oxidation, and are taken up by macrophages, which consequently transition to foam cells. This drives plaque formation and chronic inflammation at sites of endothelial damage. Atherosclerosis is, the leading cause of cardiovascular disease (CVD) [16,36,37], which itself is the leading cause of death globally (WHO Global Health Observatory). In patients with elevated blood cholesterol, interventions aim to lower LDL-C to <1.4 mmol/L (<55 mg/dL). The European Society for Cardiology guidelines suggest that blood cholesterol should be lowered by 50% with a baseline of 1.8 mmol/L–3.5 mmol/L [38]. 

Cholesterol is synthesized by the mevalonate, Bloch, and Kandutsch–Russell pathways [13]. These pathways are downregulated by the statin class of molecules that targets 3-hydroxy-3-methyl-glutaryl-coenzyme A reductase (HMGCR), the enzyme that catalyses a key interaction toward the start of the mevalonate pathway to reduce the rate of cholesterol biosynthesis [39]. Statins serve as the primary pharmaceutical intervention for elevated blood cholesterol [13,40,41] and have been demonstrated to be highly effective in preventing cardiovascular disease [42,43]. However, statin treatment can induce adverse effects, the most common being myositis, myopathy, and myalgia [44,45] and, less frequent, but more severe, rhabdomyolysis [46]. It has been estimated that 10–15% of patients using statin therapy experience such effects [47]. Statins are also known to have anti-inflammatory effects, though the mechanism of action is not well understood [48,49]. NLRP3, an intracellular danger-sensing complex, is implicated in how statins effect the immune system, driving pro-inflammatory responses via the IL-1beta and IL-18 pathways, both associated with coronary artery disease progression and plaque rupture [50,51,52]. 

Where statin treatments are ineffective or induce severe side-effects, alternative treatments include intestinal absorption inhibitors such as ezetimibe that reduce absorption from diet, but do not regulate cholesterol biosynthesis [53]. Phytosterols similarly block cholesterol absorption [54] and bile acid sequestrants and niacin can also be used in treatment [55,56]. LDL receptors (LDLr) are responsible for cellular uptake of LDL-C from the blood and Proprotein Convertase Subtilisin/Kexin type 9 (PCSK9) targets the LDL receptors (LDLr) for lysosomal degradation [57]. In recent years, proprotein convertase subtilisin kexin 9 (PCSK9) inhibitors have been approved as treatments by the U.S. Food and Drug Administration (FDA) and European Medicines Agency (EMA) [58]. These treatments inhibit the PCSK9 mediated degradation of LDLr, increasing the abundance of LDLr and the cellar uptake of LDL-C [57,59]. Moreover, currently being investigated are cholesterylester transfer protein (CETP) inhibitors that reduce the transfer of high density lipoprotein cholesterol (HDL-C), which itself is inversely correlated with risk, to atherogenic LDL-C [60,61], and Bempedoic acid that inhibits cholesterol biosynthesis upstream of HMGCR [62]. A schematic of how these treatments affect cholesterol metabolism is shown in Figure 1.

Mutations in LDLr, endocytic adaptor molecules such as PCSK9 and proteins involved in LDL formation such as APOB can cause familial hypercholesterolemia, a hereditary condition that elevates serum cholesterol and increases cardiovascular risk, affecting approximately one in 300 of the population [15,63,64]. 

Dysregulation of cholesterol and vitamin D metabolism is often observed with advancing age and is associated with an elevated risk of age-related diseases [30,31]. However, our understanding of the development of this dysregulation is ambiguous. Total serum cholesterol and LDL-C increase with age in both men and women up until the midpoint of life [65,66,67,68,69]. HDL-C is influenced considerably less by age [70,71]. In older people (>65 years), it has been observed that total serum cholesterol and LDL-C concentrations are lower than those in middle age and it is uncertain why this is the case. However, it has been speculated that this may be due to survival bias and decreased liver function [66]. 

At sites of atherosclerotic plaque formation, endothelial cells have been shown to have reduced telomere lengths and to show markers of cellular ageing, suggesting the presence of age-related disease [72,73]. There is also evidence for the relationship between lipid metabolism, longevity and healthspan with an association shown between health ageing, lipid profiles and genetic variants in enzymes that impact upon lipid metabolism, such as apolipoprotein E [74].

The age-related dysregulation of cholesterol metabolism is associated with multiple mechanistic changes that take place during ageing. The efficiency of cholesterol absorption increases with age, and this could be due to several factors. It is biologically plausible that there is an age-related increase in NPC1L1 a protein implicated in intestinal cholesterol absorption [75] and that there is age-related disruption of hepatic lipoprotein processing. It has been found that both the number and activity of hepatic LDLrs diminish in ageing rats and that this was associated with retarded chylomicron clearance [76]. In humans, age has been found to correlate with the residence time of LDL apoB-100 [77]. It is not known why the integrity of LDLr is compromised during ageing, but activation of the mammalian target of rapamycin (mTOR) complex may be involved [78]. It is to be expected that dysregulation of LDLr during ageing will elevate hepatic cholesterol levels and that this will result in a rise in plasma cholesterol levels. Other ageing processes impinging on cholesterol metabolism include oxidative damage via reactive oxygen species (ROS), which has been widely hypothesised to have a key role in ageing and age-related pathology [79]. Several lines of evidence substantiate the view that ROS are involved in cholesterol metabolism. Tentative empirical evidence from rodent studies suggest that an age-related increase in hepatic levels of ROS can provoke a rise in cholesterol biosynthesis, although recent computational work was unable to confirm this [67]. 

## 3. Vitamin D in Health and Disease

The active form of vitamin D (calcitriol, 1,25(OH)2D) is synthesised in a sequence of metabolic steps that span extracellular regions, the cytosol, lysosome, endoplasmic reticulum and mitochondria [2,80]. The metabolic pathway comprises the mevalonate pathway and the Kandutsch–Russell branch of the sterol pathways. However, in parallel with the final step of cholesterol synthesis in the Kandutsch–Russell pathway, UV-radiation regulates the transition of 7-dehydrocholesterol to pre-vitamin D3, which then isomerises to form cholecalciferol. Cholecalciferol transforms to calcidiol (25(OH)D), which subsequently transforms to the active form calcitriol (1,25(OH)2D) [2,81]. Calcitriol circulates after binding to vitamin D binding protein (GC) and dissociates in tissues to bind to vitamin D receptor (VDR) [82], its nuclear receptor transcription factor, from where it maintains calcium and phosphorous homeostatis [83]. However, it also has roles in energy metabolism, cell proliferation, differentiation, and apoptosis, and vitamin D-binding proteins in other cellular compartments have been suggested [84].

Vitamin D has been found to affect inflammation and immunity, regulating the production of inflammatory cytokines and inhibiting the proliferation of proinflammatory cells [85]. In particular, vitamin D receptor (VDR) signalling has been shown to inhibit the NLRP3 mediated immune response, which has been implicated in the anti-inflammatory effects of statins [86]. 

The definitions and thresholds for vitamin D deficiency lack a universal consensus [87]. Calcidiol (25(OH)D) is taken as a proxy biomarker of vitamin D status [88]. Deficiency has variously been defined as serum calcidiol below 25 nmol/L [89], below 30 nmol/L [82] or below 50 nmol/L [90]. Sufficiency of calcidiol (25(OH)D) has been defined at 50 nmol/L [91] and optimal calcidiol (25(OH)D) at 80 nmol/L [91,92]. In response, the National Institutes of Health established the vitamin D standardisation program (VDSP) [93,94], defining deficiency as below 50 nmol/L [95]. In the UK, the Scientific Advisory Committee on Nutrition established 25 nmol/l as a national target [96], whilst the European Food Safety Authority adopted 50 nmol/L [97]. However, it is estimated that 7% and 26% of the population of the USA show vitamin D deficiency and that 13% and 40% of the population of Europe show deficiency on average across the year, when it is defined as <30 nmol/L and <50 nmol/L, respectively [98].

Vitamin D deficiency is more common in aged populations [99,100] with estimates of 50+% of the community dwelling elderly in the U.S. being deficient [100]. Ageing leads to reduced levels of 7-dehydrocholesterol in the skin [101] and decreased expression of vitamin D receptors [102,103], along with reduced calcium absorption in the gut [1]. Muscle function has been shown to be impaired in elderly osteoporosis patients and vitamin D status has been associated with the risk of falls [104,105]. 

Vitamin D deficiency is associated with osteomalacia and rickets, amongst older and younger populations, respectively [3]. It is implicated in many non-communicable conditions including cancer, autoimmune disorders, diabetes, thyroid disorders, and cardiovascular disease [106,107,108,109], making supplementation a public health strategy [110]. Supplementation can be with ergocalciferol (a form of Vitamin D2), cholecalciferol, alfacalcidol (1α-hydroxyvitamin D), calcitriol (1,25(OH)2D) [111,112], and calcidiol (25(OH)D) [113].

Genetic variants have been identified that are associated with vitamin D deficiency including loss of function mutations in the vitamin D receptor [114,115] in CYP2R1, which converts cholecalciferol to calcidiol, and in Vitamin D Binding Protein (GC) that complexes with calcidiol, chaperoning it between compartments [116]. 

Unfortunately, clinical studies of vitamin D status can be unclear on whether the measurements taken are of calcidiol (25(OH)D), the inactive form, or calcitriol (1,25(OH)D), the active form. Similarly, studies can be unclear on which metabolite is being provided in supplementation and these factors affect our ability to explore mechanisms. However, the majority of studies that do specify use calcidiol (25(OH)D) as a biomarker. 

## 4. Computational Modelling of Vitamin D and Cholesterol Metabolism

Computational modelling provides a useful framework for studying the intersection between cholesterol and vitamin D metabolism [117]. Such models are core to the systems biology paradigm, the aim of which is to investigate biological systems in an integrated and quantitative manner. Many different theoretical approaches can be used to model a biological system. Stochastic modelling enables a system to be examined at a molecular level that captures Brownian dynamics or statistical uncertainty [118]. Macroscopic modelling can employ ordinary differential equation (ODE) based [119] and spatial ODE based [120] modelling and modal networks of interactions lend themselves to logic based modelling [121]. Metabolic systems typically suit ODE modelling due to their high particle number and the weak modality of their behaviour. 

Elements of vitamin D metabolism have been modelled including analysis of the pathway transition between calcidiol (25(OH)D) and calcitriol (1,25(OH)2D). In particular, the role of vitamin D binding protein (GC) [122,123] and calcitriol (1,25(OH)2D) have featured in a model of calcium homeostasis that promotes calcium retention [124,125]. Although not at a pathway level, the pharmacokinetics of calcidiol (25(OH)D) in HIV patients have been modelled showing that antiretroviral drugs have no effect on vitamin D status [126]. However, despite the clear clinical significance of vitamin D metabolism, the pathways involved have received relatively little attention at a systems biology level and this is reflected in the lack, at the present time, of a dedicated model of vitamin D metabolism in the Biomodels repository [127]. 

More effort has been directed towards cholesterol metabolism and the cholesterol biosynthesis pathway has been modelled at a range of levels of detail [13,41,67,128,129,130]. Additionally modelled are Sterol Regulatory Element Binding Protein (SREBP) mediated feedback [131], atherosclerosis [16,132,133,134], and cholesterol metabolism [30,41,68,130]. The potential for new pharmaceutical therapeutic strategies that target the cholesterol biosynthesis pathway has also been modelled [41,128]. However, at the present time, no work has modelled the interaction between the two pathway systems.

The challenges of computationally modelling these pathway systems are related to the paucity of parameter values that exist in the literature and online databases [128]. It has been shown that even for the cholesterol biosynthesis pathway, which is well established as critical to cardiovascular health and the target of clinically and commercially important therapeutics, only approximately half the necessary parameter values have been determined, even after pooling across relevant mammalian species [128]. 

## 5. A Bidirectional Relationship between Cholesterol and Vitamin D Metabolisms 

Vitamin D deficiency is associated with an increased incidence of CVD [21,22]. Vitamin D supplementation has been shown to improve several proxy markers of cardiovascular health [135] including lipid profiles for calcidiol (25(OHD) [23]. Calcidiol (25(OH)D) deficiency has been shown to be associated with lower HDL-C and elevated total cholesterol, LDL-C, and triglycerides with one study reporting changes of −5.1%, +9.4%, +13.5%, and +26.4%, respectively [23]. Vitamin D supplementation has been shown to lower total cholesterol, LDL-C, HDL-C, and triglycerides with standardised mean differences of −0.17, −0.12, −0.19, and −0.12, respectively, in patient cohorts [136,137]. However, paradoxically, randomised controlled trials have consistently found no reduction in cardiovascular events associated with vitamin D supplementation, though supplementation appears to reduce the mortality associated with other diseases [21,24,25,26,112,136,138,139]. 

These findings are unexpected when the role of calcitriol (1,25(OH)D) is considered. Synthesised from calcidiol (25(OHD) in the extra-renal locations of cardiomyocytes, ventricular myocardium and fibroblasts [140], calcitriol (1,25(OH)D) is involved in both cardiac remodelling and the regulation of the inflammatory processes that drive atherosclerosis including smooth muscle cell proliferation, which stabilises plaques [141]. It has been demonstrated experimentally that calcitriol (1,25(OH)D) hinders cholesterol uptake by macrophages and promotes cholesterol efflux, suggesting that vitamin D metabolites may suppress foam cell formation and therefore atherosclerosis itself [142]. In addition, it has been demonstrated that calcitriol (1,25(OHD), but not calcidiol (25(OH)D), are inversely correlated with coronary plaque burden in psoriasis patients [143]. 

The catalysis of cholesta-5,7-dien-3β-ol (7-dehydroCHOL) to cholecalciferol (VD3) in the epidermis requires UV radiation of wavelength 280–320 nm [144], although UV absorption can be affected by season, latitude, and lifestyle factors [145,146,147]. A longitudinal study showed correlation between increasing serum calcidiol (25(OH)D) and lower serum LDL-C in summer for healthy children across a wide latitude range [148]. The dependency of calcidiol (25(OH)D) on adequate sun exposure has highlighted the relationship between latitude and vitamin D deficiency [149] and between latitude and total cholesterol, LDL-C, and coronary heart disease [150,151,152]. 

## 6. The Effect of Statins

Statins target the interaction catalysed by HMGCR and suppress flux through the mevalonate and Kandutsch–Russell pathways. Statins are the primary pharmaceutical therapy for elevated blood cholesterol, and it has been estimated that in 10,000 patients treated for five years, they will prevent major vascular events in 1000 patients with pre-existing CVD and 500 events in patients without [153]. However, a small proportion of patients experience adverse effects associated with statin treatment. Amongst 10,000 patients treated for five years, statin treatment will cause ~5 cases of myopathy, 50–100 cases of diabetes, and 5–10 strokes [153,154,155]. 

Statin induced myopathy and myalgia have been shown to be associated with vitamin D deficiency and it has been established that these side-effects can be mitigated with vitamin D supplementation [28,44,45,156,157]. 

Interestingly, vitamin D supplementation in patients receiving atorvastatin treatment has been shown to lower total cholesterol and LDL-C further than atorvastatin alone. Supplementation has been shown to reduce total cholesterol and LDL-C by 12 and 14 mg/dL, respectively, over six weeks [29] and 26.0 and 22.6 mg/dL, respectively, over six months [158]. Conversely, when the effect of statin treatment on vitamin D metabolism was studied, it was initially reported that statin treatment lowers cholesterol, but raises vitamin D, in particular calcidiol (25(OH)D) [156,159,160]. However, this result has subsequently been rigorously refuted, and it is now understood that there is no or a negligible impact on vitamin D [27,161]. 

It is also valuable to consider the effect of vitamin D metabolites on statin activity. Calcitriol (1,25(OH)D) is known to induce the enzyme CYP3A4, which metabolises simvastatin, atorvastatin, and lovastatin, and the enzyme CYP2C9, which metabolises fluvastatin and pitavastatin [162,163]. Hence, calcitriol can influence the duration of action of statins. Also, the effect of atorvastatin has been shown to be weaker in vitamin D deficiency [164].

## 7. Feedback from Vitamin D Metabolites

Negative feedback in the mevalonate and Kandutsch–Russell pathways by cholesterol and its derivatives is well established [165,166,167]. However, less well known is how vitamin D regulates these pathways. It has been shown in vivo that vitamin D deficiency leads to reduced vitamin D receptor (VDR) activity and that this inhibits INSIG-2 expression, which in turn releases SREBP to upregulate the mevalonate and Kandutsch–Russell pathways [168]. A dose response suppression has been demonstrated in vitro between calcitriol (1,25(OH)D) and total cholesterol [168]. Furthermore, it has also been shown that the INSIG-2 promotor contains a response element for VDR [169]. It has been observed in vivo, that calcitriol (1,25(OH)D) and total cholesterol are inversely proportional [170], although is worth noting that this relationship is not as strong in the plasma as in the liver [170]. 

A separate feedback mechanism appears to be provided by calcidiol (25(OH)D), which has been shown to inhibit the function of HMGCR [171]. In cultured human lymphocytes, it was shown that calcidiol (25(OH)D) inhibited HMGCR activity by 63% and 93% at concentrations of 5 and 25 ug/mL, respectively. However, calcitriol (1,25(OH)D) only inhibited HMGCR activity by 20% at both concentrations, suggesting that calcidiol (25(OH)D) inhibition of HMGCR activity is independent of INSIG-2/SREBP regulation [171]. The precise mechanism by which HMGCR is inhibited is unclear. 

A further mode of cholesterol regulation appears with the activation of VDR, which has been shown in vivo to increase the activity of CYP7A1, the enzyme responsible for converting cholesterol to 7a-hydroxy cholesterol, a precursor of bile acids [170,172]. The activity of CYP7A1 undergoes regulatory feedback to modulate bile acid metabolism. Bile acids activate Farnesoid X Receptor (FXR), which in turn upregulates small heterodimer partner (SHP) to suppress CYP7A1 expression and inhibit the conversion of cholesterol to 7a-hydroxy cholesterol [173]. It has been demonstrated in vivo that calcitriol (1,25(OH)D) activation of VDR represses SHP (in a manner independent of FXR), consequently upregulating CYP7A1 expression [172]. This yields a further mechanism through which calcitriol (1,25(OHD) administration can lead to reduced serum cholesterol. 

## 8. DHCR7 and Smith–Lemli–Opitz Syndrome

Cholesta-5,7-dien-3β-ol (7-dehydroCHOL) is the metabolite at the fork in the Kandutsch–Russell pathway that contributes to both cholesterol and vitamin D biosynthesis. The enzyme DHCR7 is responsible for its conversion to cholesterol. Changes to the activity of DHCR7 can affect the balance of flux directed towards cholesterol biosynthesis and towards vitamin D biosynthesis. Low DHCR7 activity leads to the accumulation of Cholesta-5,7-dien-3β-ol (7-dehydroCHOL), which increases its rate of consumption on the vitamin D pathway [174]. DHCR7 is transcriptionally regulated by SREBP as part of the feedback that ensures homeostasis and this has been observed in statin treatment alongside proteasomal degradation in response to cholesterol surpluses [175]. 

Rare genetic diseases can serve as a clinical model for elucidation of metabolic pathway regulation. Smith–Lemli–Opitz syndrome (SLOS, OMIM #270400) is a developmental disorder caused by mutations to the DHCR7 gene that impair its function. It affects ~1:40,000 of the population [176] and leads to an accumulation of Cholesta-5,7-dien-3β-ol (7-dehydroCHOL) and a deficiency of cholesterol and its derivatives. Cholesta-5,7-dien-3β-ol (7-dehydroCHOL) itself is known to reduce the activity of the enzyme HMGCR, compounding the suppression of cholesterol synthesis [177] as well as to be a precursor of highly reactive metabolites [178]. Impaired DHCR7 activity suggests an increase in vitamin D biosynthesis and elevated vitamin D metabolites, and this has been observed clinically in SLOS patients [179]. 

## 9. Variants and Mutations

Genome Wide Association Studies (GWAS) have identified DHCR7 as a locus contributing to both cholesterol and vitamin D status [180,181]. As a result of its role in SLOS, DHCR7 has been well studied and currently, the CLINVAR database [182] lists 445 variants in DHCR7 with 168 classified as pathogenic or likely pathogenic.

The pathways leading to cholesterol and vitamin D biosynthesis are typically described as starting with acetyl coenzyme A (AC-CoA) in the cytosol. However, acetyl coenzyme A (AC-CoA) is itself formed in a sequence of metabolic steps employing a tetramer of acyl-coA dehydrogenase short/branched chain (ACADSB) [183,184]. GWAS has identified SNPs of ACADSB associated with vitamin D status and it is to be expected that the same SNPs would affect cholesterol biosynthesis, although this has yet to be studied [181]. CYP2R1 and CYP24A1 have also been identified as loci for SNPs affecting vitamin D circulation [81,180,181]. 

## 10. The Molecular Pathway of Vitamin D and Cholesterol Metabolism 

The pathway system that leads to the biosynthesis of vitamin D and cholesterol is shown in Figure 2 using Systems Biology Graphical Notation (SBGN), a standardised system of symbols for pathway maps [185]. This map is available from the Appendix A in a machine readable, semantically meaningful form using the Systems Biology Graphical Notation Markup Language format (SBGNML) [186]. It was compiled from the Reactome pathway database [187] and spans several cellular compartments and the extracellular space. The pathway starts in the cytosol where acetyl coenzyme A (AC-CoA) is catalysed by the acetyl-coenzyme A acetyltransferase 2 (ACAT2) tetramer. After a further catalysed step, the pathway enters the endoplasmic reticulum where 3-hydroxy-3-methylglutaryl-CoA (bHMG-CoA) is transformed to mevalonic acid (MVA) in a step catalysed by HMGCR and targeted therapeutically by statins in the treatment of hypercholesterolaemia [41]. The pathway returns to the cytosol briefly before continuing in the endoplasmic reticulum. Once the metabolite zymosterol is formed (ZYMOL), the pathway branches into the Bloch pathway, which includes desmosterol as a metabolite (DESMOL), and the Kandutsch–Russell pathway, which includes lathosterol (LTHSOL). Both contribute to the formation of cholesterol (CHOL) in the endoplasmic reticulum, but the Kandutsch–Russell itself also branches, with the non-cholesterol forming branch leading to vitamin D biosynthesis. The vitamin D forming branch is initiated by the transition of cholesta-5,7-dien-3β-ol (7-dehydroCHOL) to cholecalciferol (VD3), which is catalysed by UV radiation in the epidermis and, to a lesser extent, dermis [101]. The pathway leaves the endoplasmic reticulum, briefly returning to form calcidiol (25(OH)D), which is subsequently active in the lysosome, later transforming to calcitriol (1,25(OH)2D), the active form, in the mitochondria. These reactions occur predominantly in the kidney, but also in prostate, breast, and skin tissue [2,81].

Negative feedback occurs through regulation of the complexes of SREBP [166]. Binding of cholesterol to SREBP complexes leads to SREBP retention in the endoplasmic reticulum [167]. However, at low cholesterol concentrations, SREBP complexes are freer to undertake a series of catalysed steps that release SREBP multimers as transcription factors capable of upregulating the enzymes responsible for catalysing some of the early steps of the pathway [165]. 

Calcitriol (1,25(OH)2D) is subject to feedback regulation with an abundance leading to inhibition of CYP27B1, the enzyme that converts calcidiol (25(OH)D) to calcitriol (1,25(OH)2D), and stimulation of CYP24A1 (an enzyme not shown that consumes calcitriol (1,25(OH)2D), transforming it to other metabolic forms) [2]. This feedback is mediated by fibroblast growth factor 23 (FGF23) and parathyroid hormone (PTH) (not shown) [2,81].

As shown in Figure 2, calcitriol (1,25(OH)2D) drives negative feedback in the pathway, binding to VDR to activate the expression of INSIG, which in turn increases the retention of SREBP in the endoplasmic reticulum [168], and de-repressing CYP7A1 expression by inhibiting SHP, to increase cholesterol consumption [170,172]. Calcidiol (25(OH)D) is also shown to inhibit the activity of HMGCR.

## 11. Discussion

Some of the observed interplay between vitamin D metabolism and cholesterol metabolism can be explained by the feedback mechanisms and interactions described. The three mechanisms: (i) calcitriol (1,25(OH)D) driving INSIG/SREBP mediated feedback [168]; (ii) calcidiol (25(OH)D) suppressing HMGCR activity [171]; and (iii) VDR inducing CYP7A1 activity [170,172] are all consistent with the observation that vitamin D deficiency elevates total cholesterol and LDL-C and with the observation that vitamin D supplementation suppresses total cholesterol and LDL-C. 

Similarly, observations of vitamin D status, total cholesterol, LDL-C, and latitude [150,151] are consistent with UV exposure regulating vitamin D metabolites, which in turn use (i) INSIG/SREBP mediated feedback; (ii) calcidiol (25(OH)D) suppression of HMGCR; and (iii) VDR induced CYP7A1 activity to regulate the mevalonate and Kandutsch–Russell pathways and cholesterol consumption. The observation that vitamin D supplementation with statin treatment lowers cholesterol and its derivatives to levels below that observed in statin treatment alone is also consistent with the action of these three mechanisms. However, because calcitriol (1,25(OH)D) also upregulates the enzymes that metabolise statins, the effect on HMGCR may represent a balance of factors between calcidiol (25(OH)D) suppression of HMGCR and calcitriol (1,25(OH)D) induced acceleration of statin degradation [162,163,188]. 

It is unclear how vitamin D supplementation mitigates the side-effects of statin treatment, although this is largely because the side-effects themselves are poorly understood [189]. It has been proposed that the side-effects of statin treatment are at least in part triggered by deficiency of the metabolite Coenzyme Q10, which exploits HMGCR in its formation [189,190]. However, this explanation goes against our current understanding, which would predict that vitamin D supplementation reduces the activity of HMGCR, further suppressing the formation of coenzyme Q10, and not increasing it. Calcitriol (1,25(OH)D) induced accelerated degradation of statin would be consistent with this hypothesis as this would make more HMGCR available. However, it would also undermine the therapeutic value of the treatment.

It is also not clear how statin treatment succeeds in suppressing cholesterol and its derivatives, but not the vitamin D metabolites. For vitamin D biosynthesis to remain unaffected by statin treatment, the reduction in pathway flux along the Kandutsch–Russell pathway would have to be restored on the vitamin D branch by shunting flux away from the cholesterol forming branch. This could be achieved with a statin associated reduction in the activity of DHCR7. However, there is little evidence of such a link. Instead, it has been reported that cholesterol degrades the DHCR7 protein, suggesting that statin treatment succeeds in propagating DHCR7, rather than suppressing it [175]. 

The final question is perhaps the most important. When there is strong evidence that vitamin D supplementation improves important risk factors for CVD such as LDL-C, why does vitamin D supplementation fail to show improvement to cardiovascular outcomes in clinical trials? This may be linked to the observation that calcitriol (1,25(OH)D) and total cholesterol are not as strongly inversely proportional in the plasma as in the liver [170]. Cholesterol in the plasma is a risk factor for atherosclerosis, and intriguingly, it is known that, in addition to regulating the enzymes of the biosynthesis pathway, SREBP also regulates LDLr [191]. Additionally, it has been shown that calcitriol (1,25(OH)D) treatment can significantly upregulate ABCA1, the protein responsible for cellular cholesterol efflux [192]. Hence, it may be the case that vitamin D treatment downregulates cholesterol biosynthesis, but also simultaneously upregulates cellular efflux and downregulates cellular uptake of cholesterol. This would have the effect of enhancing the reduction in cellular cholesterol but working against any reduction in the plasma cholesterol. However, this is unlikely to be the whole picture. Calcitriol (1,25(OH)D) has also been shown to induce mir-1228 [193] and it has been suggested that mir-1228 targets PCSK9 [194] as well as LDLr (https://www.genecards.org/cgi-bin/carddisp.pl?gene=LDLR accessed on 2 July 2021). Together, this would suggest a rich interplay between vitamin D metabolism and the factors that affect the balance between cellular and serum cholesterol. 

## 12. Conclusions

Vitamin D metabolism and cholesterol metabolism have a complex bidirectional relationship. Certain behaviours of the two metabolisms are consistent with our understanding of the pathways and their regulation. The evidence that increased cardiovascular risk is associated with vitamin D deficiency and that sunlight exposure is associated with lower cardiovascular risk are in qualitative agreement with our understanding of the feedback and regulatory mechanisms involved. However, at present, it is not clear how vitamin D supplementation mitigates the side-effects of statin therapy, and critically, why supplementation fails to improve cardiovascular outcomes, despite improving the biomarkers of CVD risk. Here, we present a hypothesis that may explain the latter, with vitamin D regulating the balance between cellular and plasma cholesterol. Filling the gaps in our understanding of the rich interplay between these two metabolic systems has the potential to make a dramatic contribution to our knowledge of cardiovascular health and our approaches to therapy.

## Figures and Tables

**Figure 1 cells-10-02007-f001:**
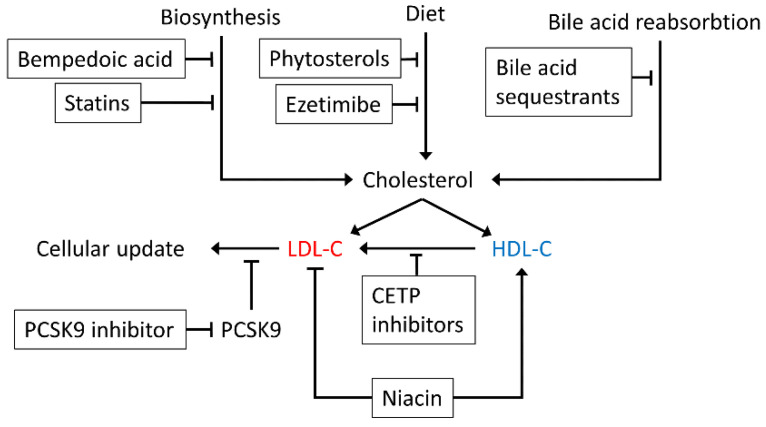
A schematic of the various medications that can be used to lower LDL-C (atherogenic, in red) and raise HDL-C (atheroprotective, in blue).

**Figure 2 cells-10-02007-f002:**
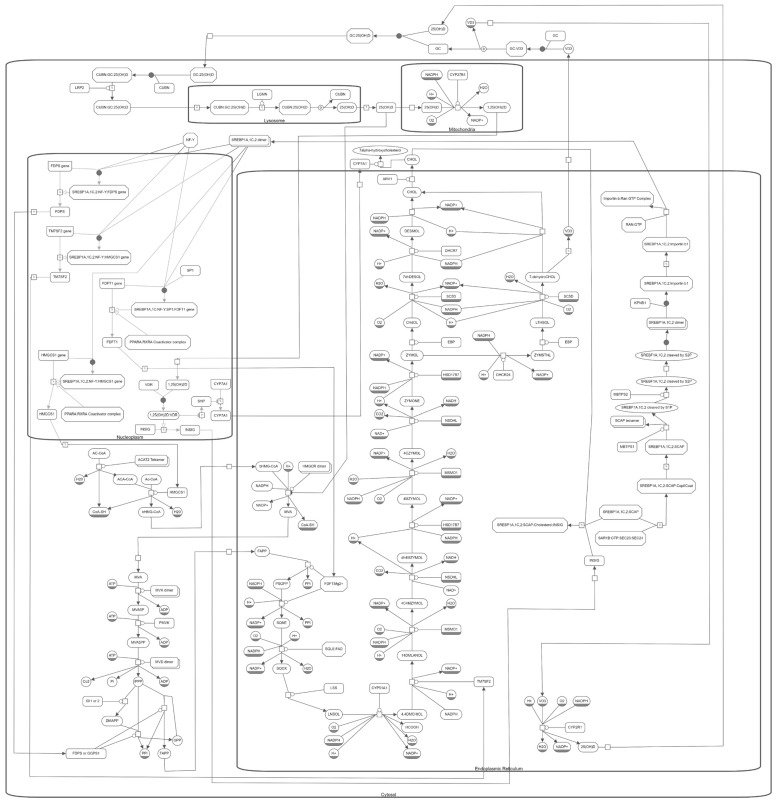
The cholesterol and vitamin D biosynthesis pathways as described in the Reactome database [187], along with the three mechanisms of feedback from the vitamin D pathway that regulate the pathways. The pathways are shown using the Systems Biology Graphical Notation [185] and this map is available for download from the Appendix A in the semantically meaningful, machine readable Systems Biology Graphical Notation Markup Language format [186].

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
