# Peer review of "The Interdependency and Co-Regulation of the Vitamin D and Cholesterol Metabolism"

_cells, 2021, doi:10.3390/cells10082007_

Round 1
Reviewer 1 Report
The authors here provide the combined metabolic pathway and clinical impact of vitamin D and cholesterol biosynthesis. Overall, this review is well summarized and written and contains valuable information to understand metabolic pathways of vitamin D and cholesterol biosynthesis and its clinical impact; however needs some improvements described below.
1. page 3 line 114 iL-1beta -> IL-1beta
2. If authors show cartoon as a figure about page3, line 116-131, it would be helpful for the reader
3. page 4 line155 “increases increase with age” should be edited
4. Higher resolution of Figure 1 would be helpful for the reader
5. If the authors provide a simple table or figure about clinical association including retrospective clinical observation of vitamin D and cholesterol biosynthesis that explained in section “Cholesterol in health and disease” and “Vitamin D in health and disease”
Author Response
We would like to thank Reviewer 1 for their suggestions and for identifying our typographic errors. Specific responses to Reviewer 1's points:-
- page 3 line 114 iL-1beta -> IL-1beta
This has been amended.
- If authors show cartoon as a figure about page3, line 116-131, it would be helpful for the reader
A schematic showing how these medications affect cholesterol metabolism is now shown in Figure 1.
- page 4 line155 “increases increase with age” should be edited
This has been amended.
- Higher resolution of Figure 1 would be helpful for the reader.
A higher resolution file of the figure is supplied.
- If the authors provide a simple table or figure about clinical association including retrospective clinical observation of vitamin D and cholesterol biosynthesis that explained in section “Cholesterol in health and disease” and “Vitamin D in health and disease”
We are a little confused about how to address this. The two sections contain descriptions of a broad diversity of observations. It is not clear how we could bring these together in a simple table or figure. The free text format allows us to write up these observations, but if we were to tabulate them, we feel that we would arrive at a list of statements from the free text as the observations aren't amenable to comparison. Similarly, because the observations cover many topics, it would be hard to capture them in a single, simple image. If guidance was provided, we would be happy to consider how to proceed with this.
Reviewer 2 Report
This is an interesting and well-articulated review article, which I feel will be of interest to researchers in the field. However, the section on cholesterol metabolism seems to be less precise than those dealing with vitamin D metabolism. The following comments apply:
- The central role of the liver in cholesterol metabolism needs to be highlighted - these comments only arise later in this section
- Cholesterol does not 'combine with LDL' - this is a misrepresentation of how LDL is formed (metabolism of VLDL)
- The role of LDL in driving macrophage 'foam cell' formation is not particularly accurate - the process is more complex than described, and is not mediated purely by phagocytosis
- The adverse impact of statins is over-stated at their first introduction - these drugs are taken by millions worldwide.
- CETP inhibitors: these are not designed to lower LDL-C, but to raise HDL-C; CETP facilitates transfer of cholesteryl ester and triacylglycerol
- ApoB100 is not a transfer protein - it is an integral apolipoprotein around which the lipids in VLDL are packaged (and thus remains with the LDL after hydrolysis of the triacylglycerol core of VLDL)
Author Response
We would like to thank Reviewer 2 for their feedback which undoubtedly removes some ambiguity from our phrasing. We have addressed the points made by Reviewer 2 as follows:-
This is an interesting and well-articulated review article, which I feel will be of interest to researchers in the field. However, the section on cholesterol metabolism seems to be less precise than those dealing with vitamin D metabolism. The following comments apply:
- The central role of the liver in cholesterol metabolism needs to be highlighted - these comments only arise later in this section
This centrality of the liver is now introduced at the start of the section, “Cholesterol in health and disease”.
- Cholesterol does not 'combine with LDL' - this is a misrepresentation of how LDL is formed (metabolism of VLDL)
The text has been modified to be more clear.
- The role of LDL in driving macrophage 'foam cell' formation is not particularly accurate - the process is more complex than described, and is not mediated purely by phagocytosis
The text has been modified to be more clear.
- The adverse impact of statins is over-stated at their first introduction - these drugs are taken by millions worldwide.
The text has been modified to describe the proportion of cases in which adverse effects occur.
- CETP inhibitors: these are not designed to lower LDL-C, but to raise HDL-C; CETP facilitates transfer of cholesteryl ester and triacylglycerol
The text has been modified to be more clear. Additionally Reviewer 1 has requested a schematic that also describes the role of CETP inhibitors.
- ApoB100 is not a transfer protein - it is an integral apolipoprotein around which the lipids in VLDL are packaged (and thus remains with the LDL after hydrolysis of the triacylglycerol core of VLDL)
The text has been modified to be more clear.
Reviewer 3 Report
The authors have presented a review article to discuss the interplay and co-regulation of vitamin D and cholesterol metabolism, the understanding of both is important for cardiovascular and other diseases. They have discussed the role of both chemicals individually and combined in human health and disease, the effect of statins on both pathways, genetic variants and feedback loops in these pathways. It also remains a conundrum as to why vitamin D supplementation, despite being a potent anti-inflammatory fails to have any cardiovascular benefit. This important point is discussed here.
This is a well-written review article of moderate importance in the field. I do have a few major comments and suggestions to improve the article.
Major
1-The major novelty is the first and very extensive systems biology pathway map of joint cholesterol and vitamin D metabolisms, but the figure resolution is too small to be visible.
2- The line on page 2, "..cardiovascular events were ultimately unaffected by (vitamin D) supplementation..." is important. This discussion is also found on page 6. However, the flaw in all of these epidemiological studies is the assumption (by Cashman (cited here)) that inactive 25OHD is the proxy marker of vitamin D. The authors should cite and discuss the paper by Playford MP et al. Atherosclerosis in 2019 which compares the serum levels of 25OHD and 1,25 OHD2 and cardiovascular risk. The levels of 1,25 are inversely associated with CV risk but 25OHD are not suggesting that 1,25 should be considered when assessing the role of vitamin D on cardiovascular disease.
3- It is more preferable to cite high quality peer reviewed original research articles rather than obscure review articles to support the discussion. For example, are there alternative citations that can be used for
a. Igbal et al. to support that statins don't influence plasma vitamin D levels?
b. Kim et al. 2019 and Khouski K, 2021 to discuss that statins may have anti-inflammatory effects (this is a questionable suggestion anyway)?
c. Bhattacharyya et al. 2012 to assume that calcitriol can influence the duration and action of statins?
4. Many other biological processes diminish with advancing age. Thus, it is not surprising that deregulation in both vitamin D and cholesterol metabolism similarly reduces with age. I'd recommend removing these paragraphs on page 3 and 4. Maybe replace with a more extensive discussion of vitamin D on cholesterol efflux?
Minor
1- When describing "vitamin D" in the article, please accurately determine which form. For example on page 2, ...."does not influence plasma levels of vitamin D". Is it 25OHD or 1, 25OHD2? Also on page 7, "...statin treatment lowers cholesterol, but raises vitamin D."
2- Why is inactive 25OHD and not 1,25 OHD2 more potent in inhibiting HMGCR activity? Page 7.
Author Response
We would like to thank Reviewer 3 for their feedback and their specific comments around the ambiguity of vitamin D labelling. This is something that we have wrestled with in writing the review as not all studies report the metabolites they use. We have addressed their points as follows:-
The authors have presented a review article to discuss the interplay and co-regulation of vitamin D and cholesterol metabolism, the understanding of both is important for cardiovascular and other diseases. They have discussed the role of both chemicals individually and combined in human health and disease, the effect of statins on both pathways, genetic variants and feedback loops in these pathways. It also remains a conundrum as to why vitamin D supplementation, despite being a potent anti-inflammatory fails to have any cardiovascular benefit. This important point is discussed here.
This is a well-written review article of moderate importance in the field. I do have a few major comments and suggestions to improve the article.
Major
1-The major novelty is the first and very extensive systems biology pathway map of joint cholesterol and vitamin D metabolisms, but the figure resolution is too small to be visible.
A higher resolution file of the figure is supplied.
2- The line on page 2, "..cardiovascular events were ultimately unaffected by (vitamin D) supplementation..." is important. This discussion is also found on page 6. However, the flaw in all of these epidemiological studies is the assumption (by Cashman (cited here)) that inactive 25OHD is the proxy marker of vitamin D. The authors should cite and discuss the paper by Playford MP et al. Atherosclerosis in 2019 which compares the serum levels of 25OHD and 1,25 OHD2 and cardiovascular risk. The levels of 1,25 are inversely associated with CV risk but 25OHD are not suggesting that 1,25 should be considered when assessing the role of vitamin D on cardiovascular disease.
We have now referred to the work by Playford et al. in the section, "A bidirectional relationship between cholesterol and vitamin D metabolisms". The results we have presented predominantly agree with the findings of Playford et al. showing that it is 1,25(OH)D that is active in CYP7A1 processing of cholesterol and in INSIG/SREBP mediated feedback in the biosynthesis pathway.
3- It is more preferable to cite high quality peer reviewed original research articles rather than obscure review articles to support the discussion. For example, are there alternative citations that can be used for
- Igbal et al. to support that statins don't influence plasma vitamin D levels?
Iqbal et al. is not a review article and though the journal is not high impact, the study is listed on Pubmed. We feel uncomfortable subjectively dismissing the study as obscure. Sahebkar et al., Aloia et al. and Perez-Castrillon et al. all support the same result and we have added a further reference to a study (Gupta and Thompson).
- Kim et al. 2019 and Khouski K, 2021 to discuss that statins may have anti-inflammatory effects (this is a questionable suggestion anyway)?
We agree that the use of original research would be more helpful to the reader here. We have added citations to original research (Satoh et al.), but retained the citation to review articles as we feel that they neatly summarize a range of results immediately relevant to the topic. The anti inflammatory effects of statins were also covered in a Nature Review on the topic (Greenwood et al) which we have now included.
- Bhattacharyya et al. 2012 to assume that calcitriol can influence the duration and action of statins?
We have now cited the original research (Thummel et al., and Drocourt et al.)
- Many other biological processes diminish with advancing age. Thus, it is not surprising that deregulation in both vitamin D and cholesterol metabolism similarly reduces with age. I'd recommend removing these paragraphs on page 3 and 4. Maybe replace with a more extensive discussion of vitamin D on cholesterol efflux?
This manuscript was submitted to a special issue on “Mechanisms of Aging and Therapeutic Approaches to Target Age-Associated Chronic Diseases” and we felt that those paragraphs helped to establish the relevance of vitamin D and cholesterol metabolism to the theme of the special issue. We are happy to accept guidance from the editor on this matter and remove the paragraphs if the editor thinks that it is appropriate.
Minor
1- When describing "vitamin D" in the article, please accurately determine which form. For example on page 2, ...."does not influence plasma levels of vitamin D". Is it 25OHD or 1, 25OHD2? Also on page 7, "...statin treatment lowers cholesterol, but raises vitamin D."
We have added a paragraph to the section "Vitamin D in health and disease" describing the lack of specificity in the literature on this topic as it is not always reported which metabolite is supplied/measured. Additionally we have been more specific in key places in the manuscript (line 285, line 337). However, in other places, such as the introduction, pg 2, we felt it appropriate to write about supplementation in general and more abstract terms.
2- Why is inactive 25OHD and not 1,25 OHD2 more potent in inhibiting HMGCR activity? Page 7.
This is unknown as far as we have been able to determine from the literature. We have added a lined to state this (line 365).
Round 2
Reviewer 2 Report
The authors have addressed the misleading comments in the original version